# Survey of Emotional Themes Used in Marketing of Commercial Baby Foods in the UK—Implications for Nutrition Promotion in Early Childhood

**DOI:** 10.3390/ijerph21030258

**Published:** 2024-02-23

**Authors:** Ada Lizbeth Garcia, Nicole Chee, Elisa Joan Vargas-Garcia, Alison Parrett

**Affiliations:** Human Nutrition, School of Medicine, Dentistry & Nursing, University of Glasgow, Glasgow G31 2ER, UK; cheew.nick@yahoo.com (N.C.); elisa.vargas08@gmail.com (E.J.V.-G.); alison.parrett@glasgow.ac.uk (A.P.)

**Keywords:** infant feeding, nutrition, baby foods, commercial determinants of diet

## Abstract

Claims used in the marketing of commercial baby foods are often misleading, and there are concerns that they exploit parental anxieties. We adapted a hierarchical consumer emotions model to explore the emotional themes used in the marketing of commercial baby foods sold in the UK market. A survey administered in three large UK supermarkets collected in-store data on commercial baby food characteristics and the marketing claims used on commercial baby food packaging. The keywords found in these claims were entered in N-Vivo and allocated to four preexisting emotional themes: contentment, happiness, love, and pride. The prevalence of each theme was compared by age suitability (4+, 6+, 9+, and 12+ months) and taste (sweet/savoury) profile. A total of 1666 marketing claims (median 5, IQR 3) and 1003 emotional keywords (median 3, IQR 3) were identified on the packaging of 341 commercial baby foods. Foods suitable for infants aged 6+ months displayed more claims (50%, *p* < 0.05) and emotional keywords (56%, *p* = 0.07). Savoury foods displayed more emotional keywords (63%, *p* < 0.001). The keywords “little”, “encourage”, “love(ly)”, and “tiny” were the most frequently used words under the theme of love (36% total contribution). The emotional connotations of the keywords under the theme “love” are extensively used in the marketing claims on commercial baby food packaging. These might exploit parental vulnerabilities and influence their purchasing of commercial baby foods.

## 1. Introduction

Optimal nutrition during the first year of life is essential for growth and development, but also to promote healthy eating and adequate feeding behaviours in later childhood and to reduce the risk of chronic disease later in life. The current global recommendations for feeding children during their first year are to exclusively breastfeed from birth for up to 6 months and then to introduce complementary foods to meet the child’s increasing energy and nutrient requirements [1]. The quality of complementary foods is of great importance because children have a limited gastric capacity and are at risk of energy and nutritional imbalances if not supplied with nutrient-dense foods [2]. Foods containing critical nutrients, such as protein, essential fatty acids, calcium, iron, zinc, vitamin B12, vitamin A, and iodine, should be prioritised over foods with lower nutrient density [1]. Nutritional imbalances have negative implications for morbidity and increase the risk of mortality in at-risk populations. Complementary foods that are high in sugar, salt, and trans and saturated fats should not be given to babies because they displace other nutrient-dense foods. It is also essential to start promoting healthy eating habits early in life [1]. Complementary foods play a role in shaping infants’ development of food preferences [3]. The degree to which infants are exposed to a variety of flavours and textures during complementary feeding influences food intake in later childhood [4] and can extend into adulthood [5]. It is well established that humans have an innate preference for sweet foods, and therefore these foods are easily consumed by young children [6]. Thus, it is important to expose an infant’s palate to a wider variety of flavours whilst limiting sugar intake. This is important for good oral health and for weight gain prevention [2]. On the other hand, infants reject bitter and sour tastes as an innate biological response, and therefore they should be exposed to these flavours to promote preferences for vegetables and fruits [7]. Low fruit and vegetable consumption, particularly bitter types, is a concern because these are sources of vitamins, fibre, and other bioactive compounds needed to establish immune responses and a healthy gut microbiome and for the amelioration of inflammatory processes, which are commonly present during childhood [8]. Infants also need to develop feeding skills by learning how to feed themselves. This can be achieved by offering finger foods that are suitable in terms of texture and size. Whole finger-size pieces of fruits, vegetables, cheese, and bread are among the foods recommended for self-feeding [2].

Commercial baby foods are discouraged because of concerns about their nutrient content, ingredient composition, flavour/taste, texture, and limited variety. Several studies in low-, middle-, and high-income countries have found that a large proportion of commercial baby foods have high sugar and sodium content and are low in iron [9,10,11,12,13,14,15]. We have previously reported that the ingredients used in the formulation of commercial baby foods are predominantly sweet. For example, they often combine cereals or animal sources of protein (e.g., milk, cheese, chicken) with sweet types of vegetables (e.g., sweet potato, carrot) and fruit (pear, apple, banana) [16]. This is problematic as sweetness is the predominant taste that children will be exposed to if frequently fed commercial baby foods. The textures of commercial baby foods are also questionable; a large proportion of these foods are soft and come in the form of purees offered in pouches [14]. Harder textures are available in the form of baby snacks, but these are designed to be crunchy and easily melt in the mouth [17], giving them an organoleptic appeal that mimics snacks for older ages. The lack of adequate textures is relevant because infants need to learn how to chew and bite foods to stimulate their oral muscle development and their chewing and swallowing skills [18]. Commercial baby snacks have grown in availability [17], and thus there is a growing concern that snacking on highly processed foods will become a normalised behaviour, but this is to be discouraged. The consumption of foods low in nutrients and high in fat, sugar, and salt in infancy is associated with the double burden of malnutrition in low- and middle- income countries [19,20] and with higher energy intake and weight-for-length z scores in more affluent countries [21]. Other aspects of concern include the price of commercial baby foods; this is often higher compared to homemade foods [15]. Nonetheless, a large proportion of infants are offered commercial baby foods on a regular basis; for example, in Scotland, 41% of mothers of 8–12-month-old infants reported feeding them commercial baby foods on five or more days per week, and 24% stated that they gave their children “treats” as a snack at least once a day (these included chocolate buttons, ice cream, crisps, and cheese puffs) [22].

Given the popularity of commercial baby foods and the concerns around the health implications of using commercial baby foods, it is also essential to consider the role of marketing and the tactics used by the food industry to position and sell their products. The baby food industry is a multinational business that uses aggressive marketing strategies that jeopardise the health of infants [23,24]. The messages used in the marketing of commercial baby foods contradict infant feeding recommendations; although improvements have been made concerning the age range it is suggested the food product is suitable for, companies still display 4 instead of 6 months as a suitable age to start complementary feeding. This diminishes the exclusive 6 month breastfeeding recommendations and confuses parents, leaving them unsure about which advice to follow [17]. Food labels often display misleading names for the ingredients used in their formulations. For example, food names like “spinach, apple, and swede” suggest that a green vegetable is the main ingredient, but fruit is the main ingredient [25,26]. Commercial baby snacks use messages related to self-feeding behaviour, such as “encourages self-feeding” and “ideal finger food” [25], which promote snacking on highly processed foods [27]. Our group and others have reported the extensive and pervasive use of marketing claims on the packaging of commercial baby foods. In our study of 734 products, we identified 6265 promotional claims on packaging (an average of nine per product). Almost all of the products included a marketing claim; common claims were related to the composition and nutrient content, whilst direct health claims were uncommon [25]. The use of a “healthy halo” thematic, which is misleading, is a common practice by the baby food industry [25,28,29,30].

Given all these issues, the World Health Organization (WHO) has suggested that the inappropriate promotion of commercially available complementary foods for infants should be stopped. This includes the abolition of marketing claims, including those idealising a product, as well as misleading health and compositional claims [31]. Some countries have already taken regulatory action to minimise or even prohibit the use of manipulative marketing for products targeted at children [32]. However, fewer actions have been taken for commercial baby foods. This is important because, in marketing, brand loyalty results from autobiographical memory and habituation [33]. Attributes such as “attractive brand promotion” in commercial infant foods contribute to parental decisions to choose specific food brands [34]. Parental decisions on infant feeding during the first year are important to drive feeding trajectories in childhood and beyond. For example, feeding babies formula milk is associated with not following healthful complementary feeding recommendations, including consuming more treats and commercial baby foods [35]. The consumption of commercially available, highly processed foods in pre-school years is associated with higher energy intake and adiposity in later childhood [36].

The WHO Europe has proposed a tool to monitor the nutrient content of commercial baby foods in order to end the promotion of foods that do not meet nutrient and marketing recommendations. This is a comprehensive tool; however, its implementation is voluntary, which may be a barrier to its use. Other technical barriers to the implementation of this tool include a lack of portion size standardisation, a lack of detailed information on free sugar content in food tables, the difficulty in setting sugar threshold limits for fruit-based foods, the consideration of protein and iron limits, and uncertainty regarding the acceptable limits on the use of sodium for technological purposes [37]. In terms of ending the promotion of inappropriate foods, the marketing aspect of the tool focuses on nutrient profiling, baby food names, and claims [23]. We argue that there are other aspects of marketing that are not included in the proposed framework. These are related to the idealisation of products and social desirability (e.g., the promotion of dessert-type or snack foods as a social norm). Such aspects are often used in the promotion of commercial baby foods by displaying messages conveying positive experiences, such as fun, heritage, problem solving, and happiness [28,30]. A more in-depth understanding of the use of emotional themes that appeal to parents’ feelings is needed. This is important because parental choice in the purchasing and use of commercial baby foods is influenced by emotional traits, such as increased anxiety due to factors including food safety (e.g., fear of choking and gagging while self-feeding) and a lack of confidence during meal preparation. It has been argued that food companies use and reinforce these arguments via marketing [38]. We also hypothesise that sweet baby foods, which are predominant in the baby food market, are more likely to use emotional themes. This is based on our previous observations that marketing claims are more prevalent in sweet snacks [25]. We aim to explore the emotional themes used on the packaging of commercial baby foods in the UK and how these vary by recommended age and taste profile.

## 2. Materials and Methods

### 2.1. Study Design

This was an exploratory survey of commercial baby foods sold in stores in the UK.

We adapted an existing framework to classify emotional themes. The framework was based on a hierarchal approach to emotions in consumer behaviour described by Laros and Steenkamp [39]. Under this framework, emotions are classified into three levels: superordinate, basic, and subordinate. The superordinate level distinguishes emotions into positive and negative affects only. The basic level is defined as emotions experienced at an “innate and universal” level; these are sub-classified into positive (contentment, happiness, love, and pride) and negative (anger, fear, sadness, and shame). Finally, the subordinate level consists of more specific emotions, such as guilt, relief, or warm-heartedness. For this study, only the positive affects at the superordinate and basic levels were adopted. Negative affects were excluded because we hypothesised that the marketing claims would focus on positive emotions when promoting the consumer purchase of their products [3]. The subordinate level was also omitted to avoid unnecessary complexities and allow this modified framework to be comprehensible and easily adapted for other types of products. The four basic categories of positive affect were maintained. For the purpose of our study, these categories were defined as (i) contentment: passive positive emotions, low arousal, reassuring; (ii) happiness: reactive positive emotion, high arousal; (iii) love: interpersonal emotion, endearment, nurture; and (iv) pride: feeling of superiority, e.g., “my baby is developing faster”.

### 2.2. Data Collection

Data were collected from 3 major supermarket chains in Glasgow, United Kingdom, with approximately 46% of the market share (Tesco, Morrisons, Aldi) [28]. Data were collected in-store by photographing all sides of the packaging of the commercial baby foods available in the baby food aisle of these supermarkets in the period of September to December 2021.

Variables recorded were the brand, name of product, age group, taste profile (sweet, savoury), and marketing claims of the commercial baby foods. Only commercial baby foods advertised as being suitable for children aged 4–12 months were included and were categorised as 4+, 6+, 9+, and 12+ months. Taste profiles were classified following our previous commercial baby food survey methodologies [2,6]. Briefly, sweet commercial baby foods were “fruit-based meals, deserts, puddings, rusks, dairy products (e.g., custard, yogurt), or fruit-containing foods such as breakfast cereals and dry snacks”, while savoury commercial baby foods consisted of “meat, poultry, fish, vegetables, cheese, pulses, or carbohydrate-based foods (including savoury snacks), or a combination of these”. A classification of food categories using the Nutrient Profile Model (NPM) [37] was also applied (Table A1). The type of marketing claims found on any side of the food packaging was entered and classified according to the categories described in our previous survey. These were texture, baby-led weaning, convenience, conveying ideals on optimal feeding, dietary goals, endorsements, lifestyle, quality, taste, and others [21]. These claims were further used to extract emotional keywords.

#### Analysis and Categorisation of Consumer Emotional Behaviour

Emotional keywords that were regularly used in the surveyed marketing claims on commercial baby foods were entered using the word frequency function in NVivo 12 (Version 12.4.0.741) 64-bit (2020). These keywords were categorised by one researcher according to Laros and Steenkamp’s framework [39] and were reviewed by another researcher for accuracy. The research team built upon the framework by bringing additional marketing claims into the four basic categories. For instance, claims such as “perfect”, “balance(d)”, and “complete” were placed under the category of contentment, on the assumption that commercial baby foods displaying these terms were marketed as superior in comparison to homemade meals. Commercial baby foods are often marketed according to age or “stage”, which ranges from 1 to 4. The stages are not standardised as children reach feeding milestones at different times; however, most commercial baby foods follow these conventions. Children aged 4–6 months are in stage 1, children aged 6–9 months are in stage 2, children aged 10–12 months are in stage 3, and children aged over 12 months are in stage 4. As the stages progress, the texture and taste of these commercial baby foods are made to be more complex. Thus, stage is classified under the category “pride”, on the assumption that if the parent feeds their child commercial baby foods in a higher stage than their age implies, this may encourage them to believe that their child is surpassing others. Similarly, “different” or “however” is categorised under the category “pride” only if the word “baby” is also present; an example of this is “Government guidelines advises weaning from 6 months. However, every baby is different”. These marketing claims may suggest that the consumer’s child is more advanced in comparison to others, while undermining feeding guidelines.

In the instance that multiple keywords were identified in a single marketing claim, each keyword was recorded and assigned in accordance with its emotional theme. The categorisation of emotional keywords into themes was done by two members of the research team (N.C. and E.J.V-G.), with a third member (A.L.G) participating in the discussions.

### 2.3. Data Analysis

Descriptive statistics were analysed using Microsoft Excel version 2111 (2021) and IBM SPSS version 28.0 (IBM corp, 2021). Chi-square tests were used to test associations between age recommendations and taste profiles and emotional themes, as well as for food categories, taste profiles, and the use of emotional themes. The significance level was set at *p* < 0.05.

## 3. Results

### 3.1. Commercial Baby Food Characteristics and Marketing Claims

A total of 342 commercial baby foods from 13 individual brands were identified in stores during the data collection period. A total of 1663 marketing claims were identified. Texture-related claims were the most prevalent (63%), followed by baby-led weaning (20%) and quality (12%) (Figure 1). Commercial baby foods displayed a median of five (IQR 3) claims per product. Examples of marketing claims by type included texture (e.g., smooth blend, puffed, soft, textured to encourage chewing, crunch-tastic, crispy, designed to encourage your baby to chew, lip smacking); baby-led weaning (e.g., finger food, encourages self-feeding, ideal finger food, chunky puffs for little fingers); and quality (e.g., nutrition for little tummies, personally guaranteed, contains real fruit). Endorsement (e.g., nutritionist-approved) and quality (e.g., made with organic rice and natural flavours, baked with baby grade ingredients) were also common. 

The general characteristics of commercial baby foods are shown in Table 1. A larger proportion of commercial baby foods are recommended for those aged 6+ months. The proportions of sweet and savoury foods were similar. Age recommendations were associated with the number of marketing claims displayed, with a greater proportion of claims found in commercial baby foods recommended for infants aged 6+ months. Age recommendations and taste profiles were also associated with the number of emotional keywords used in commercial baby foods: higher proportions of emotional keywords were observed in foods suitable for those aged 6+ months (19%, *p* < 0.05) and in savoury foods (63%, *p* < 0.001). Food categories stratified by taste profile and emotional keyword in commercial baby foods are shown in Table A1. Only seven categories were inputted as no drinks were recorded during data collection. Most foods under the sweet category were fruit and vegetable purees, whilst savoury meals were the most common under savoury taste. Emotional words were more likely to be displayed on savoury meals, followed by snacks/finger foods, which are mostly of a savoury taste profile.

### 3.2. Emotional Keywords and Themes

A total of 1003 emotional keywords were identified. The median number (IQR) of emotional keywords per product was three (IQR 3). Table 2 shows the keywords allocated to the four basic emotional themes in commercial baby foods. Keywords such as “little, encourage, love (ly), and tiny” were the most frequently used words under the theme of love. Examples of claims under the theme of love include “made with love”, “loved by parents and babies for generations”, and “for lively little learners”. Keywords such as “yummy, explore, adventure, and packed” were often used under the theme of happiness; examples of phrases under this theme are “made for playing & learning”, “a world of adventure”, and “yummy organic food”. Top keywords under the theme of contentment were “perfect, balance (d), and easy/ease”. Examples of claims using the word “perfect” are “perfect first foods” and “I am perfect for lunchboxes and picnics”. Keywords related to age recommendations (stage), “different (baby)”, and approved were the most found under the theme of pride; examples for this theme include “nutritionist approved” and “Ingredients suitable from 4 months. Government guidelines advise weaning from 6 months. Every baby is different”!

## 4. Discussion

The marketing strategies used by the commercial baby food industry have been under scrutiny due to concerns about the nutritional quality of commercial baby foods, the expansion of the commercial baby food market, the pervasive nature of the marketing that occurs, and the lack of regulations [25,30,40]. The WHO Europe has recently proposed a tool to monitor the nutrient profiles and marketing of commercial baby foods, but this has not been implemented yet [23]. This tool is a comprehensive guide to establish nutrient composition and marketing claims, but we believe that the marketing claim component is mostly focused on nutrient and health claims. Other promotional messages that idealise food products are featured to a lesser extent. An area of concern that has received little attention in the evidence about inappropriate commercial baby food marketing is the use of words and phrases that exploit parental anxieties and aspirations [30,38]. This is a similar tactic to those used by the formula milk industry. Such tactics are applied to packaging and promotions and include segmenting parents based on parenting style (“future ambitions and aspirations for their children”, “assurance of current happiness”, and “protective parents”) [41,42].

Feeding infants during the complementary feeding period can be an overwhelming task if parents lack confidence, have limited food literacy, and/or lack time [43,44]. Parents’ information sources on infant feeding include information provided by commercial baby food companies [43,45]. Thus, it is important to understand which messages are being conveyed to parents and how this is being achieved.

We set out to examine whether the age and taste profiles of commercial baby foods were associated with emotional claims and found that, in contrast to our hypothesis, savoury foods, particularly those under the categories of savoury meals and snacks/finger foods, were more likely to display emotional keywords. This could be due to the increased popularity of snacks for self-feeding and the promotion of snacking behaviours using claims related to texture [17]. Another reason for this could be that companies are shifting from the promotion of sweet foods to savoury foods due to concerns about the sugar content and the sweet taste of commercial baby foods [46]. Furthermore, the WHO Europe tool has set up stringent thresholds for sugar content in commercial baby foods to be <5%; this will make it difficult for sweet products to be classified as suitable for infants [23]. A further reason could be the appeal of promoting vegetables as part of a savoury food. Commercial baby snacks are often marketed as vegetable-based, e.g., “carrot sticks”, “tomato and basil puffs”, and “pea puffs”, but this is misleading because it implies that the snacks are composed of these vegetables.

We identified a large number of marketing claims, comparable with our report in 2022 [25]. In this study, we explored other dimensions of the marketing claims and found that keywords under the theme “love” were the most prevalent, which is not surprising, given the intrinsic drivers of parenthood and duty of care, which are influenced by affection. Parents are motivated to offer their child the best start in life, including to support optimal health and development [47]. For example, “breast is best” is a message that must be included in formula advertising, which is then used by infant formula companies together with the word “however” to further introduce reasons to explain why formula is important [48]. This is hugely problematic because it diminishes the breastfeeding recommendations. Similar tactics are used in the promotion of commercial baby foods. For example, due to increased scrutiny regarding the recommended age to introduce complementary foods to align with the WHO recommendation to start complementary foods at 6 months, baby food companies acknowledge these recommendations by displaying the message “Government guidelines advises weaning from 6 months”, but they also use the word “however” to add keywords linked to the theme of pride, e.g., “every baby is different”. Such marketing messages are confusing for parents [44] and can exploit the lack of parental knowledge around when it is best to introduce complementary foods.

Messages that appeal to parental emotions are widely used in marketing claims. This provides further evidence on another dimension of the commercial determinants of infant feeding and the need to fully understand and act upon the tactics used by baby food companies [16]. The food environment plays a key role in shaping parental feeding decisions in a period in which parents are vulnerable and need support. Parents’ decisions on infant feeding are complex; they are influenced by socio-political, socio-economic, and socio-environmental factors [49]. Thus, solutions that empower parents should be multi-layered. Supporting food environments that promote health in the early years must include legal frameworks and allow monitoring and sanctioning mechanisms [50]. This translates into greater control over how commercial baby foods are marketed and extends beyond current practices, i.e., food composition and nutrient and health claims. Measures to target emotional advertising are needed. Inappropriate labelling preys on parents’ vulnerabilities, manipulating them to feel as if they are expressing love, when in fact these actions might unknowingly result in lifelong unhealthy taste preferences. Therefore, the reformation of marketing claim regulations on commercial baby food is necessary to develop a healthier future generation. Our study provides initial evidence of the extensive use of emotional themes, which fall under “other tactics”, used by the baby food industry to promote a commercial influence on infant feeding. This has been well characterised in the marketing of breast milk substitutes and should be considered in future regulation and policy for commercial baby food [42].

Other tactics that the food industry has used to market unhealthy foods aimed at children include—among others—the use of child imagery, child-friendly language, and bright attractive colours on food packaging and in retail marketing, e.g., positioning and special offers. The WHO has called for stricter policies to regulate harmful marketing aimed at children; these policies have the potential to reduce purchasing and exposure [51]. In this study, we did not collect specific information on the use of child imagery on baby food packaging, but this was among the criteria for data inclusion.

Further exploration of how parents interpret emotional messages will be needed to provide guidelines for public health practice and policy. More research is needed to investigate the effect that these subtle but emotional messages have on parental purchasing decisions; this will provide evidence to develop nutrition promotion actions for parents to raise awareness of feeding recommendations and the ways in which the infant food industry manipulates parental anxieties.

### Strengths and Limitations

The strength of this study is the comprehensive analysis of emotional keywords displayed on all the baby foods surveyed in the store. This provides insights into the messages used to promote other positive attributes of commercial baby foods, which are merely subjective. The categorisation of the emotional themes was done by two members of the research team, with a third member participating in the discussions.

The main limitation of the study is its cross-sectional nature, which constrained its time validity. The data were collected in 2021. We reviewed the availability of the products obtainable online in the same stores during the first two weeks of December 2023 and found that 73% were still available, and the names of the products were the same. This shows the rapid changes in product availability in the baby food market. Another limitation was that we did not survey all the main supermarkets in the UK; however, those that we surveyed represented 46% of the market share and several baby food companies sell the same products in all stores. Commercial baby foods might also be available online and we did not include these during the data collection in 2021. In our 2020 survey, we explored food claims on 720 commercial baby foods in seven UK supermarkets and online on Amazon. Here, we report claims from only half of this number of products. We might have missed keywords from products, which calls for a further in-depth analysis of a larger number of foods in future studies. Furthermore, a more objective method to determine whether the words used in the classification are indeed emotionally appealing to parents needs to be investigated. This modified framework can be further tested with parents.

## 5. Conclusions

Marketing claims in commercial baby foods include an extensive number of keywords with emotional connotations. Keywords under the theme of “love” were often used, which might exploit parental vulnerabilities to influence their purchasing and use of commercial baby foods. Given the extensive use of emotional themes in the marketing of commercial baby foods, it is imperative to explore how these affect parental food purchasing and feeding decisions and to include these aspects in future commercial baby food labelling regulations and marketing policies.

## Figures and Tables

**Figure 1 ijerph-21-00258-f001:**
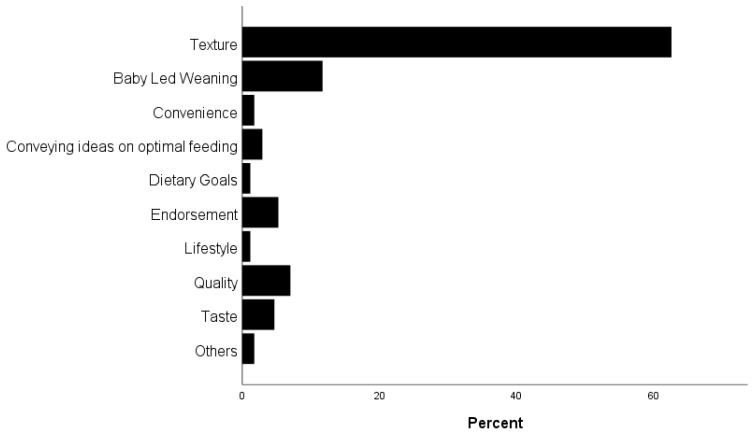
Distribution of marketing claims by type.

**Table 1 ijerph-21-00258-t001:** Characteristics of commercial baby foods, marketing claims, and emotional keywords.

Characteristics	General		Marketing Claims		*p* Value	Emotional Keywords		*p* Value
	N	%	N	%		N	%	
Age								
4+	63	18	317	19		145	14	
6+	173	51	832	50		557	56	
9+	40	12	207	12		139	14	
12+	65	19	310	19		162	16	
Total	341	100	1666	100	<0.05	1003	100	0.07
Taste								
Sweet	162	48	753	45		371	37	
Savoury	179	52	913	55		632	63	
Total	341	100	1666	100	0.165	1003	100	<0.001

**Table 2 ijerph-21-00258-t002:** Keywords allocated to emotional themes in commercial baby foods.

Emotional Theme	Proportion of Emotional Theme (%)	Keywords	Keyword Frequency	Weighted Percentage (%)
Contentment	26	Perfect	110	10.90
		Balance(d)	53	5.25
		Easy/Ease	31	3.07
		(No/Less) Mess	15	1.49
		Support	16	1.59
		Guarantee	12	1.19
		Trust	12	1.19
		Complete	9	0.89
Happiness	28	Yummy	84	8.33
		Explore	58	5.75
		Adventure	41	4.06
		Packed	24	2.38
		Play	21	2.08
		Discover	16	1.59
		Fun	11	1.09
		Burst	9	0.89
		Scrummy	9	0.89
		(No/Less) Nasties	6	0.59
		Exotic	3	0.30
		Tickle	1	0.10
Love	36	Little	83	8.23
		Encourage	66	6.54
		Love(ly)	47	4.66
		Tiny	46	4.56
		Develop	27	2.68
		Learn	24	2.38
		Family	21	2.08
		Home	11	1.09
		Care	9	0.89
		Mini	8	0.79
		Nurture	6	0.59
		Small	6	0.59
		Tummy	6	0.59
		Dinky	5	0.50
Pride	10	Stage (Numbered)	35	3.48
		Different (Baby)	29	2.87
		Approved	25	2.48
		Award/Winner	6	0.59
		However (Baby)	2	0.20
Total:	100		1003	100.00

## Data Availability

Data are available upon reasonable request.

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
