# Peer review of "Survey of Emotional Themes Used in Marketing of Commercial Baby Foods in the UK—Implications for Nutrition Promotion in Early Childhood"

_ijerph, 2024, doi:10.3390/ijerph21030258_

Round 1

Reviewer 1 Report

Comments and Suggestions for Authors

Dear authors, 

This is a great research paper which captures the tactics used by industry to influence purchasing behaviors. Please see below comments for your consideration.

Introduction/Discussion - include additional content regarding the power of and exposure to marketing on purchase behaviors and long-term impacts.

Line 111 - WHO tool limitations - could you clearly unpack the limitations of the tool, please check the reference provided.

Line 182 - Could information on the product categories (using the WHO Europe NPM 2023 or UK 2004 to 2005 NPM) of the 342 baby foods be provided as an annex or within a table of results.  Is there an association/pattern between emotional key words/themes and particular categories? What % of the foods will be products in scope for the UK advertising restrictions on TV and online for products high in fat, salt and sugar? 

Line 283 - Exploiting parents, see WHO report: how the marketing of formula milk influences our decision on infant feeding (this includes data from the UK as well as other countries) and Breastfeeding 2023 Lancet Series for additional evidence. 

Line 290 - Comparison between the previous report in 2022 and the data provided in the paper, have there been changes in words, approaches, and/or Trade names for the products?

Line 329 - did this research collect information on the use of cartoon mascots on HFSS foods? Provide additional context to the tactics of product packing, and retail marketing, only claims has been discussed thus far. Consider referencing the WHO systematic review and meta-analysis on the impacts of food marketing on children's diet-related outcomes for marketing on food packaging.

Line 350 - if the reported data was collected in 2021. What year was the data collected for the 2022 report - is the report published?

Discussion - how should this research inform the UK trading standards.

References - reference numbering appears not to match between in text referencing and the reference list. Also review the style guide for references.

Author Response

Thank you for your comments, it has been extremely valuable to have your review. We have answered all questions and addressed your suggestions as best as we could. Please find the response in the attached document. 

Reviewer 2 Report

Comments and Suggestions for Authors

Reviewer notes

Title: Emotional themes used in marketing of commercial baby foods in the UK – implications for nutrition promotion in early childhood

-        It helps the reader if authors consider improving the title to reflect the study methodology as well.

General comments:

-        Authors explored a not so-often addressed dimensions of marketing claims (i.e., claims related to parental emotions) on commercial baby food products in UK, having greater potential to exploit parental vulnerabilities and influence purchase decision makings. This is an area that needs to be explored more to inform and strengthen nutrition policies regulating commercial baby food products in the region and elsewhere. This reviewer felt authors were a bit too soft on their concluding remarks given the novelty of their study [see more details on this below]. The manuscript generally reads well and communicates the essence of the study to the reader. Some minor comments and improvement ideas are presented below for the authors consideration:

Abstract

-          It helps the reader if authors clearly indicate the study design in the abstract.

-          The implication of the study could be improved: “…exploit parental vulnerability and influence purchase of commercial baby food” –what does this mean? How does this affect nutrition promotion in early childhood? What’s are the nutrition policy implications for early childhood feeding practices? Authors need to identify some policy implications to better align their conclusions with the study title.         

Introduction

-          Authors have done a good job in provides comprehensive background to the issue at hand and the rationale for the current study.

Methodology

-          This section provides description of how commercial baby food claims are categorized across four basic emotional themes, but authors fail to provide information on how baby foods are categorized across flavor or texture profiles and the rationale for doing so.

Results

-          Well presented…

Discussion, strength/limitations & conclusions

-  Discussions are well written. Strength and limitations are sufficiently highlighted.

Conclusions are too soft. Notwithstanding the limitations described, authors could make their conclusions stronger.  The mechanisms of how these parental-emotion-appealing claims on baby food products might influence purchase decisions are well described. Early childhood nutrition policy implications of such emotional dimensions of marketing claims are described. However, as mentioned in my comment under the abstract section, these implications are not clearly highlighted in the authors concluding remarks which misses a huge opportunity to highlight the potential implication of their work.  

Author Response

We thank you for taking the time to review our paper. We found your comments very useful and have addressed your questions and suggestions as best as we could. Please find attached the response to your review in a separate file. 
